**Original research**

# Active travel to school: a longitudinal millennium cohort study of schooling outcomes

Ian Walker,[1] Tim Gamble  [2]

¹School of Psychology, Swansea University, Swansea, UK
²School of Psychology, University of Surrey, Guildford, UK

**Correspondence to**
Professor Ian Walker;
ian.walker@swansea.ac.uk

## ABSTRACT

**Objectives** Assess longitudinal associations between active travel during the school commute and later educational outcomes.

**Setting** England, Wales and Northern Ireland.

**Participants** 6778 children, surveyed at ages 7, 11, 14 and 17.

**Primary and secondary outcomes** School-leaver General Certificate of Secondary Education exam scores summed to provide a single measure of educational success.

**Results** Controlling a range of sociodemographic and health variables, using active versus passive travel modes during a child's commute to school during earlier years predicted differences in school-leaver exam performance at age 16. These effects were mediated through changes in self-esteem, emotional difficulties and behavioural difficulties. Examples include: being driven to school at 11 was associated with improved exam performance at 16 mediated through enhanced self-esteem at 14 ($ab$=0.08, 95% CI=0.01 to 0.20, p=0.05) and cycling at 14 was associated with better exam scores at 16 mediated through reduced emotional difficulty at 16 ($ab$=0.10, 95% CI=0.01 to 0.30, p=0.05). The relationship between travel mode and exam performance was moderated by household income quintile, most notably with poorer exam performance seen in high-income children who were driven to school. Importantly, although our model predicted 21% of variance in exam performance, removing travel mode barely reduced its ability to predict exam scores ($\Delta R^2$=−0.005, $F_{20,6469}$ = 2.50, p<0.001).

**Conclusion** There are differences in school-leaver exam performance linked to travel mode choices earlier in the school career, but these differences are extremely small. There appears to be no realistic educational disadvantage from any given travel mode, strengthening the case for cleaner, healthier modes to become the default.

## INTRODUCTION

A child's daily journey to school provides routine opportunities for physical activity that might have important health consequences.[1–3] Active travel modes such as walking, cycling and scooting provide valuable daily exercise that is absent when children are driven to school, to say nothing of how driving children to school creates further serious public health problems through air-quality

### STRENGTHS AND LIMITATIONS OF THIS STUDY

⇒ We used the high-quality UK Millennium Cohort dataset, following a large sample of children over 10 years, enabling us to consider a broad range of health and demographic covariates.
⇒ We used official, national-standard exam results as the outcome measure.
⇒ Data on travel to school were self-reported and reduced each child to a single 'main' mode, thereby not fully capturing the details of children whose journeys were more complex.
⇒ Data collection was at 3-year intervals and so this dataset would not allow us to identify specific ages that are critical in development at anything other than a crude level.

impairment. As well as obvious barriers to the adoption of more active travel for the school commute, such as geography and societal norms, there is a further barrier arising from the informal learning opportunities linked to the daily commute.[4] It has long been claimed that children who travel to school passively—like being chauffeured in cars—can miss out on everyday learning opportunities during their commutes compared with children who travel independently,[4–6] and might also be less attentive on arrival at school than children who use active modes like walking.[3 7] This might appear to make active travel the obvious choice except, in direct contrast to the studies just mentioned, it has also been claimed that being driven to school by adult caregivers every day provides children with important educational opportunities[6 8 9] from, for example, obtaining parental advice on homework. These conflicting claims, and a lack of evidence, mean policymakers cannot know the wider effects of supporting sustainable active travel to school (as is legally mandated in countries like the UK).[10] A strong test of this issue would be to ask whether any effects of travel to school can be picked up in official end-of-school exam grades, and this is what we provide here.

Although the mechanisms by which everyday travel might affect learning at school[4–6] have been discussed since the early 1990s,[4–6] evidence in this area remains weak.[11 12] Five components of childhood well-being have been distinguished (physical, psychological, social, economic and cognitive)[13]—the last of which provides a plausible route between physical activity and school performance.[11] A recent comprehensive review[12] suggested that cognitive ability, and thereby academic attainment, might be influenced by children's experiences on the journey to school through mechanisms like increased alertness[3] in walkers, cyclists and bus users, but noted that evidence remains anecdotal or inconsistent, a point also made recently elsewhere.[11] Enhanced peer relationships are another mechanism suggested to explain how independent travel might benefit children who walk or take buses to school,[6 12] but there are also qualitative data from multiple sources suggesting that regular parental interaction for children who are escorted in cars could have educational benefits.[6 8 9] The contradictory claims in this area are impossible to resolve given the existing evidence[11 12] and so we performed a direct test of whether how children travel to school could predict their end-of-school exam scores several years later. The objective was to assess longitudinal associations between active travel during the school commute and later educational outcomes. Given the past research outlined here, we hypothesised that if we saw any educational advantages for users of certain travel modes, these might be separately mediated through changes in behavioural/attentional problems (which might plausibly be improved by regular physical activity)[3 7 11 12] and self-esteem (which might plausibly be improved from experiences of autonomy in school travel).[4 6 7 11 12]

## METHODS

We tested our hypotheses using the Millennium Cohort Study dataset,[14] a UK birth cohort study which tracks people from their birth in 2000–2001 to adulthood through multiple waves. The study used a cluster-stratified framework, oversampling groups such as those from the smaller nations of the UK, those from ethnic minority groups and those from disadvantaged areas. The entire dataset for the first wave comes from around 19 000 participants.[14] After dropouts, and after excluding cases in Scotland (where the exam system is different), cases with incomplete data on travel behaviour across ages 7, 11 and 14 and cases where travel mode was listed as 'other' or 'bike (someone else cycles)', we had records from 6778 participants who also provided data on General Certificate of Secondary Education (GCSE) or iGCSE (international GCSE) exams at age 16 (5033 from England, 1023 from Wales, 722 from Northern Ireland; 3078 male, 3423 female, 276 missing sex data). GCSEs provide a useful metric as they are intended to assess a broad range of the subjects learnt at school and a set of (typically) 7–9 GCSE assessments has been undertaken by almost everybody in England, Wales and Northern Ireland at age 16 since

the mid-1980s. We calculated a total numeric school-leaving exam score for each person using a government points system[15] to translate letter grades into numerical equivalents, giving greater credit to those who obtained higher grades and those who chose to take extra exams. The mean exam score for the sample was 46.27 points (SD=21.87) which roughly corresponds to a child obtaining 5 B grades and 4 C grades.[15]

The original dataset uses numerical codes to describe the following categories: public transport, school bus/coach, car/taxi, bike, bike (someone else cycles), walking, or refusal/don't know/not applicable/other. We reduced these to: 'Public transport', 'School bus or coach', 'Private motorised', 'Bike' and 'Walk', and excluded the other categories from analysis. At age 7, travel to school was by the following modes: 3089 walking, 108 public transport, 198 school bus, 3329 private car and 54 bicycle. At age 14, the numbers were: 2462 walking, 1152 public transport, 1207 school bus, 1832 private car and 125 bicycle. Data on who the children travelled with on their way to school were only collected at age 11. However, there were clear signs that many respondents misunderstood this question and so the data were not used (of the 11-year-olds travelling to school in cars, 2323 were recorded as travelling 'with adults' but 739 were 'with other children' and 150 'on their own'!).

We ran a series of causal mediation analyses[16] to assess plausible mechanistic pathways from travel mode to exam performance. We tested three potential mediators: Externalising, which is a collective measure of inattentiveness, hyperactivity and conduct problems from the Strengths and Difficulties Questionnaire (SDQ);[17] SDQ Internalising, which measures emotional and peer-relationship difficulties;[17] and the Rosenberg Self-Esteem measure.[18] SDQ scores were parent-reported except at ages 14 and 16, when they were child-reported. Each mediation analysis controlled for demographics and behavioural factors including sex, ethnicity, country of residence, distance from school, household income quintile, private schooling and—because physical activity was a plausible mechanism behind any travel-mode effects on educational success[3 8–10]—leisure-time sport and incidental physical activity. The Millennium Cohort contains weightings to enable descriptive statistics properly to represent the various countries within the UK, but these were not needed here given that we did not report simple descriptive statistics, and given country was included as a predictor in the analysis. The following temporal chains were assessed: travel at 7→mediators at 11→exams at 16; travel at 7→mediators at 14→exams at 16; travel at 11→mediators at 14→exams at 16 and travel at 14→mediators at 16→exams at 16. Mediation coefficients *ab* are interpreted as the mean marginal (ie, controlling for covariates) change in exam score associated with the travel mode in question which is explained specifically through changes in the mediating variable.[16] Walking was set as the reference travel mode. CIs were estimated using non-parametric bootstrapping with 500 repetitions.

**Table 1** Linear regression model predicting total school exam scores at age 16 from demographic and travel variables. Part 1: demographics

| | | Coeff. | SE | t | P value | 95% CI lower | 95% CI upper |
|---|---|---|---|---|---|---|---|
| | (Intercept) | 26.46 | 1.39 | 19.03 | <0.001 | 23.74 | 29.19 |
| Country | England | Ref | | | | | |
| | Wales | 4.25 | 0.72 | 5.92 | <0.001 | 2.84 | 5.66 |
| | Northern Ireland | 3.71 | 0.84 | 4.41 | <0.001 | 2.06 | 5.36 |
| Sex | Male | Ref | | | | | |
| | Female | 6.31 | 0.49 | 12.80 | <0.001 | 5.35 | 7.28 |
| Ethnicity | White | Ref | | | | | |
| | Mixed | 3.44 | 1.46 | 2.35 | 0.02 | 0.57 | 6.31 |
| | Indian | 8.93 | 1.41 | 6.31 | <0.001 | 6.15 | 11.70 |
| | Pakistani/Bangladeshi | 9.49 | 1.16 | 8.18 | <0.001 | 7.21 | 11.77 |
| | Black | 4.78 | 1.51 | 3.16 | 0.002 | 1.82 | 7.75 |
| | Other inc. Chinese | 16.63 | 1.93 | 8.64 | <0.001 | 12.86 | 20.41 |
| Household income quintile | 1 (lowest) | Ref | | | | | |
| | 2 | 5.09 | 1.51 | 3.37 | <0.001 | 2.13 | 8.06 |
| | 3 | 10.35 | 1.40 | 7.41 | <0.001 | 7.61 | 13.09 |
| | 4 | 15.01 | 1.37 | 10.98 | <0.001 | 12.33 | 17.69 |
| | 5 (highest) | 24.47 | 1.38 | 17.74 | <0.001 | 21.77 | 27.18 |
| Mental health issue | No | Ref | | | | | |
| | Yes | −4.39 | 1.10 | 3.99 | <0.001 | −6.55 | −2.23 |
| Longstanding illness | No | Ref | | | | | |
| | Yes | −2.83 | 0.73 | 3.88 | <0.001 | −4.25 | −1.40 |
| Private schooling | No | Ref | | | | | |
| | Yes | 2.77 | 0.93 | 2.98 | 0.003 | 0.95 | 4.58 |
| Sport at age 7 | per one point increment | 1.60 | 0.21 | 7.66 | <0.001 | 1.19 | 2.01 |
| Sport at age 11 | per one point increment | 0.99 | 0.17 | 5.77 | <0.001 | 0.65 | 1.33 |
| Other physical activity at 7 | per one point increment | −0.40 | 0.15 | 2.67 | 0.008 | −0.70 | −0.11 |
| Other physical activity at 11 | per one point increment | −0.50 | 0.14 | 3.50 | <0.001 | −0.79 | −0.22 |

Note: The travel components of this model are found in table 2. Statistically redundant predictors and interaction terms have been removed.

Linear regression analysis predicted total exam score for each child from travel behaviour at 7, 11 and 14 using the same demographic and behavioural covariates as the mediation analyses. The covariates were also tested for moderation effects with transport mode. A process of model simplification used backward deletion to remove statistically redundant predictors and interaction terms and so leave the minimal adequate model.

## Patient and public involvement

Patients and the public were not involved in this study.

## RESULTS

Our analysis found pathways between school travel mode and later exam performance, separately mediated through self-esteem, externalised behaviour problems and internalised socio-emotional difficulties. The pathways with significant indirect mediated effects were as follows: cycling at 7 was associated with lower exam performance at 16 mediated through self-esteem changes at 11 ($ab$=−0.80, 95% CI=−1.61 to −0.12, p=0.04); being driven at 11 was associated with improved exam performance at 16 mediated through enhanced self-esteem at 14 ($ab$=0.08, 95% CI=0.01 to 0.20, p=0.05); being driven at 14 was associated with better exam performance at 16 mediated through enhanced self-esteem at 16 ($ab$=0.19, 95% CI=0.08 to 0.34, p<0.001); public transport at 14 was associated with better exam scores at 16, also mediated through enhanced self-esteem at 16 ($ab$=0.18, 95% CI=0.04 to 0.36, p=0.01); cycling at 14 was associated with worse exam scores at 16 mediated through greater externalising at 16 ($ab$=−0.78, 95% CI=−1.54 to −0.06, p=0.03), although cycling at 14 was also separately associated with better exam scores at 16 mediated through reduced internalising at 16 ($ab$=0.10, 95% CI=0.01 to 0.28, p=0.05). Reduced internalising at 11 also mediated a

**Table 2** Linear regression model predicting total school exam scores at age 16 from demographic and travel variables. Part 2: travel behaviour

| | | Coeff. | SE | t | P value | 95% CI lower | 95% CI upper |
|---|---|---|---|---|---|---|---|
| Transport mode age 14 | Walking | Ref | | | | | |
| | Public transport | 2.04 | 1.92 | 1.06 | 0.29 | −1.73 | 5.81 |
| | School bus or coach | −2.54 | 2.58 | 0.99 | 0.32 | −7.59 | 2.51 |
| | Private motorised | 0.17 | 1.67 | 0.10 | 0.92 | −3.10 | 3.45 |
| | Cycling | 9.12 | 6.24 | 1.46 | 0.14 | −3.12 | 21.35 |
| Age 14 public transport (v. walking)×household income quintile 2 (v. 1) | | −1.22 | 2.73 | 0.45 | 0.65 | −6.58 | 4.13 |
| Age 14 public transport (v. walking)×household income quintile 3 (v. 1) | | −0.98 | 2.47 | 0.40 | 0.69 | −5.82 | 3.86 |
| Age 14 public transport (v. walking)×household income quintile 4 (v. 1) | | 1.11 | 2.41 | 0.46 | 0.64 | −3.60 | 5.83 |
| Age 14 public transport (v. walking)×household income quintile 5 (v. 1) | | −2.88 | 2.34 | 1.23 | 0.22 | −7.46 | 1.70 |
| Age 14 school bus (v. walking)×household income quintile 2 (v. 1) | | 5.95 | 3.21 | 1.85 | 0.06 | −0.34 | 12.24 |
| Age 14 school bus (v. walking)×household income quintile 3 (v. 1) | | 4.66 | 2.99 | 1.56 | 0.12 | −1.20 | 10.52 |
| Age 14 school bus (v. walking)×household income quintile 4 (v. 1) | | 6.50 | 2.91 | 2.24 | 0.03 | 0.81 | 12.20 |
| Age 14 school bus (v. walking)×household income quintile 5 (v. 1) | | 1.99 | 2.87 | 0.69 | 0.49 | −3.64 | 7.61 |
| Age 14 private motorised (v. walking)×household income quintile 2 (v. 1) | | 1.54 | 2.42 | 0.63 | 0.53 | −3.21 | 6.28 |
| Age 14 private motorised (v. walking)×household income quintile 3 (v. 1) | | 0.38 | 2.17 | 0.18 | 0.86 | −3.87 | 4.63 |
| Age 14 private motorised (v. walking)×household income quintile 4 (v. 1) | | 1.61 | 2.07 | 0.78 | 0.44 | −2.45 | 5.67 |
| Age 14 private motorised (v. walking)×household income quintile 5 (v. 1) | | −4.64 | 2.03 | 2.28 | 0.02 | −8.62 | −0.67 |
| Age 14 cycling (v. walking)×household income quintile 2 (v. 1) | | −16.33 | 8.45 | 1.93 | 0.05 | −32.90 | 0.23 |
| Age 14 cycling (v. walking)×household income quintile 3 (v. 1) | | −19.70 | 7.49 | 2.63 | 0.009 | −34.38 | −5.02 |
| Age 14 cycling (v. walking)×household income quintile 4 (v. 1) | | −2.22 | 7.24 | 0.31 | 0.76 | −16.43 | 11.97 |
| Age 14 cycling (v. walking)×household income quintile 5 (v. 1) | | −12.85 | 6.91 | 1.86 | 0.06 | −26.39 | 0.69 |

Note: The demographic components of this model are found in table 1. Statistically redundant predictors and interaction terms have been removed.

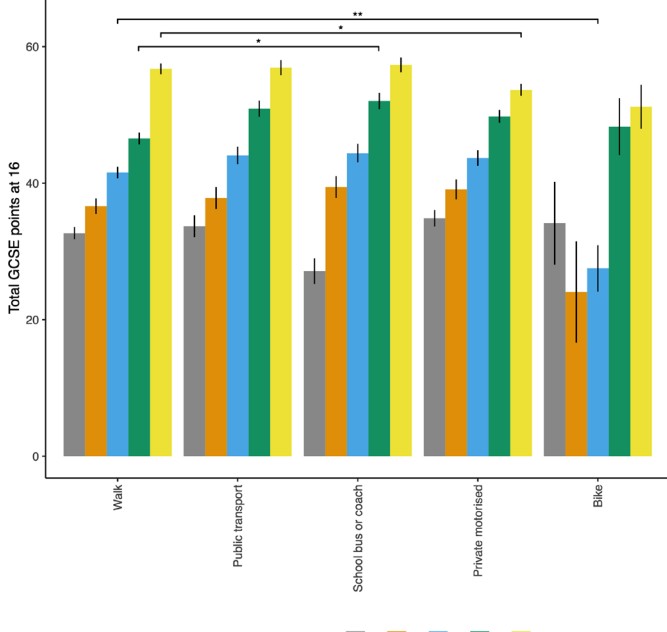

**Figure 1** School exam scores at age 16 from n=6778 children according to household income quintile and travel mode at age 14. Walking was set as the travel reference category. *p<0.05, **p<0.01. GCSE, General Certificate of Secondary Education.

relationship between public transport at 7 and increased exam scores at 16 (*ab*=0.47, 95% CI=0.02 to 0.96, p=0.04). All other mediated pathways between travel history and exam scores were non-significant (p≥0.10).

A regression analysis (reported across tables 1 and 2) found that overall exam performance at 16 was related to the travel mode used to reach school 2 years earlier, and to demographic variables including sex and household income ($R^2_{adj}$=0.21). Many of the control variable effects were expected (eg, a mean marginal exam advantage of 6.30 points enjoyed by girls),[19] which supports the validity of the analysis. All minority ethnic groups performed substantially better than the reference white category and there were very clear cumulative advantages for children in higher-income households, peaking at a marginal mean difference of 24.46 points for the highest quintile versus the lowest (roughly equivalent to eight exams each rising from grade C to grade A or to a child gaining three extra high-grade qualifications). The influence of travel mode was significantly moderated by household income quintile (figure 1); perhaps the most notable aspect of this interaction was that, in the highest household income group, there was a mean marginal reduction of 4.63 exam points (equivalent to three exams each dropping from an A to a B) for the children who were driven to school (figure 1).

Critically, however, entirely removing travel mode from our model barely changed its overall ability to predict exam scores ($\Delta R^2$=−0.005, $F_{20,6469}$=2.50, p<0.001). This reveals travel mode to be a trivial influence on exam scores compared with the demographic and behavioural

covariates. This accords with the small effect sizes described above, for example, being driven at 14 led to a mean mediated change in exam scores of just 0.20 points across all exams versus walking, which is extremely small against an overall mean exam score of 46.27 points.

## DISCUSSION

This analysis has shown for the first time that there are specific psycho-behavioural pathways linking travel to school in teenage years with later academic outcomes, as assessed through end-of-school exam results. The pathways were particularly mediated through changes in self-esteem, and to some extent behavioural and socio-emotional difficulties, that in turn were influenced by travel experiences earlier in a child's school career. Prior work has distinguished two domains that might be associated with daily travel: psychological (well-being, independence) and cognitive (mental ability, academic attainment).[11 12] The present study has demonstrated that these two domains interact such that the psychological domain might have an effect on the cognitive domain. It is notable that being driven to school by adult caregivers at 14 and independently using public transport at 14 both produced similar effects on later exam performance through a route involving enhanced self-esteem. This implies that, contrary to past claims,[4–6 8 9] it cannot simply be quality time with adults or independent mobility that uniquely boosts school success. Rather, there appears to be more than one route to better school outcomes as a result of daily travel experiences and both active and passive modes can provide some of these benefits provided they have immediate effects on mediating psychological factors like self-esteem, emotional adjustment or behavioural difficulties.

That last point notwithstanding, this study's core finding is to show, using an adequately powered sample, that while components of childhood well-being like self-esteem and socio-emotional difficulties[11–13] can be influenced by the way a child travels to school, and that while these in turn might influence exam success, the magnitude of this effect on school outcomes is dramatically smaller than the influences of demographics, health and socioeconomic status.[12] Once these covariates were controlled, the way children travelled to school explained just 0.5% of unique variance in exam outcomes. Given the size and quality of the sample used here, we can state with some confidence that the way children travel to school is a real, but very minor, influence on their scholastic success.

This study draws strength from its large, representative, high-quality longitudinal sample and its use of an official, national-standard set of examinations—taken by practically everybody in the three countries studied—as the outcome measure. Limitations attach to the self-report measures of travel mode (in particular we identified concerns with the quality of some of the data on adult escorts) and our method needing to reduce each child to a single mode, thereby likely missing nuances of

children who have more complex school journeys, such as returning home using a different mode.

## CONCLUSION

Our finding that there is no clear educational advantage for any one travel mode should end some long-standing speculation within the research literature about the impact of travel experiences on learning[3–9 11–13] and thereby help unlock public health gains by removing an impediment to reducing the use of private cars for the school commute. Modes like walking, cycling and buses are immediately positive for children's well-being[20] and also provide health benefits,[1 2] so it is perhaps surprising they do not translate into more substantial academic performance advantages. On the other hand, as we have clearly shown here, nor does spending time in cars with adult caregivers. Given the known negative consequences of driving children to school, including air pollution and lack of physical activity,[1 2 11] it is now clear that policymakers can recommend active and public transport modes as the default choice without fear of meaningful long-term educational disadvantage.

**Acknowledgements** We thank Anthony Laverty for his advice on working with the Millennium Cohort dataset.

**Contributors** IW, TG contributed to conceptualisation. IW, TG contributed to methodology. TG contributed to data management. IW contributed to statistical analysis. IW contributed to visualisation. IW contributed to writing–original draft. IW, TG contributed to writing–review & editing. Guarantor: IW.

**Funding** The authors have not declared a specific grant for this research from any funding agency in the public, commercial or not-for-profit sectors.

**Competing interests** None declared.

**Patient and public involvement** Patients and/or the public were not involved in the design, or conduct, or reporting, or dissemination plans of this research.

**Patient consent for publication** Not applicable.

**Ethics approval** This study involves human participants and was approved. As this study involved secondary analyses of data that do not contain identifiable information, the MCS ethical approval was not required. Data collection for the MCS has ethical approval from the Yorkshire and Humber ethics committee (11/YH/0203) and further details are available from http://www.cls.ioe.ac.uk/. Participants give consent for their data to be used for research purposes. Participants gave informed consent to participate in the study before taking part.

**Provenance and peer review** Not commissioned; externally peer reviewed.

**Data availability statement** Data are available in a public, open access repository. MCS data are available from https://ukdataservice.ac.uk/. Study pre-registration available from https://osf.io/u9wdr. Analysis code available from https://osf.io/46eq3.

**ORCID iD**
Tim Gamble http://orcid.org/0000-0002-6610-0835

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
