## [Reviewer comments · BMJ Open]

ARTICLE DETAILS

TITLE (PROVISIONAL)	Active travel to school: a longitudinal Millennium Cohort study of schooling outcomes
AUTHORS	Walker, Ian; Gamble, Tim

VERSION 1 – REVIEW

REVIEWER	van Sluijs, Esther MRC Epidemiology Unit
REVIEW RETURNED	08-Nov-2022

GENERAL COMMENTS	This is an interesting longitudinal study looking at association between travel modes during earlier years and later educational outcome measured by school-leaver GCSE exam scores. This study is adequately powered and included children from three different nations within the UK. This study also included mediation analyses to explain the mechanistic pathways from travel mode to exam scores. The analyses done in the study were able to address the research questions. Interpretation of the results and conclusions were appropriate. However, this manuscript was challenging for readers not familiar with the UK Millennium Cohort and the UK school system and further detail is required to enhance readability for an international audience and to understand the methods more generally. I provide more detailed comments below. Major comments Introduction 1. Objective statement should be clear in the Introduction so readers can easily match the objective and the result. It was already clear in the Abstract but was not present in the Introduction. Methods 2. Methods section should follow the STROBE guideline.3. Please add a brief description about the UK Millennium Cohort at the beginning of the Methods section to help readers who are not familiar with the cohort. For example, include details on participant recruitment and explain who you included in this study and whether inclusion/exclusion criteria were applied.4. Please explain about GCSE exam because readers not familiar with the UK school system may not understand why the exam is relevant to be used as an outcome in this study.5. Please elaborate more on the three mediators, for example how they are measured, and brief descriptions and validation information of the scales used to measure the three mediators rather than just the names of the scale.6. The UK Millennium Cohort sample is provided with weights so that it can be representative of each nation. However, it was not clear in the methods section that the sample is weighted and
--

	whether the statistical analysis took sample weighting into account. Results 7. Please include a brief description of the sample included in the study (e.g., how many were girls and boys, how many belong in certain household quintiles, many were driven by car in year 7 and how many in year 14). Discussion 8. Page 10, line 34: In here you state adequately powered sample, but it was not clear how the sampling in the UK Millennium Cohort was done. The strength of the sampling method was not clear as well. Please elaborate more on this. Minor comments: Abstract 9. The explanation about travel mode was significantly moderated by household income quintile was important and should be mentioned in the abstract. Introduction 10. Page 5, Line 34: What are these important educational opportunities? If word limit allows, please elaborate. 11. Is there any literature on benefit of walking together with parents to school? Because it seems that in the last paragraph of the introduction independent travel is pitched against escorted car travel with parents where interaction with parents becomes the latter's example of benefit. I think this interaction can also happen where children walk together with their parent/s to school. Methods 12. Page 7, line 34: I think creating a graph to show the temporal chains would be helpful to aid the readers understanding of them. 13. Please describe how travel to school mode was assessed in the UK Millennium Cohort. From the Methods section, it is not clear how many different options of travel modes are available from the methods, even though later we could clearly see in the results table. 14. Why was walking used as the reference instead of being driven by car? I would assume the narrative would be easier if being driven by car was the reference. Results 15. If possible, please create a visualisation for the results of the causal mediation analyses. Discussion 16. Page 10, line 18: Was there any instance that some children that chose walking to school were walking together with their parents to school in the UK Millennium Cohort? If yes, it can be assumed that the UK Millennium Cohort was not able to capture this. How would this fit with the quality time with adults (in car trip to school) vs independent mobility? Conclusion 1. Page 11, line 18: "it is perhaps surprising they do not translate into more substantial academic performance advantages" – do you have a research recommendation to address this? Figure 1 2. Have you checked whether the palette used to create this graph is colour blind-friendly? 3. The lines with * and ** above the bars are confusing; it is not clear which part of the text in the results it is referring to.
--	--

REVIEWER	Fyhri, Aslak Norwegian Centre for Transport Research
REVIEW RETURNED	09-Jan-2023

GENERAL COMMENTS	A well written and concise paper. Maybe a bit too concise. I would like to have some more details about the methods and some elaboration of strengths and limitations in the discussion. Importantly, the data (.i.e sample, survey instruments etc, study design) about travel mode to school, which after all is the main input variable to the analysis is not described. There is a reference to a secondary publication, where i suspect these might be better described. But that is not good enough. So please provide more info in this in the methods section. The authors acknowledge that the way in which travel mode data is captured is a limitation, but this is only in the abstract. Please elaborate briefly on the major strengths and weaknesses in the discussion. I could not see any supplementary material, so i have not been able to ascertain whether this was sufficiently completed (as per Q13 in the review form).
---

REVIEWER	Laverty, Anthony Imperial College London, Primary Care and Public Health
REVIEW RETURNED	17-Jan-2023

GENERAL COMMENTS	Thanks for sending me this paper to review. It is an interesting read and I applaud the authors for submitting this for publication even in the absence of 'large' or 'positive' results. I think it is well conducted and written; I recommend it for publication and only have very minor comments The abstract gives a nice sense of the paper. I would clarify here what scale these GCSE scores are measured in as most people are aware of these are being based on letter grades. The introduction is useful and interesting. I was not aware of the research/claims that being driven to school is an educational opportunity (!). the wellbeing components are nicely set out Methods The sample size is a bit lower than the overall N in the MCS. The missingness across waves and variables could be better described. The analyses is well motivated. One minor suggestion is about the exam score – could this perhaps be converted into standardised units or similar for use in the results and tables. This might help the reader have a sense as to the size of these associations A little more detail on how this final model was chosen would be helpful beyond – the model simplification was based on theory, or on statistical significance? The results and discussion do a nice job of setting out the main points and findings. There is not much on the limitations which could usefully be added
---

VERSION 1 – AUTHOR RESPONSE

Reviewer: 1

Major comments

Introduction

1. Objective statement should be clear in the Introduction so readers can easily match the objective and the result. It was already clear in the Abstract but was not present in the Introduction. A sentence echoing the Abstract's objective statement has now been added as the penultimate sentence in the Introduction.

Methods

2. Methods section should follow the STROBE guideline.

STROBE guidelines have been addressed in the checklist requested by the editor, as noted above.

3. Please add a brief description about the UK Millennium Cohort at the beginning of the Methods section to help readers who are not familiar with the cohort. For example, include details on participant recruitment and explain who you included in this study and whether inclusion/exclusion criteria were applied.

The description of the UK Millennium Cohort now reads: "We tested our hypotheses using the Millennium Cohort Study dataset¹⁴, a UK birth cohort study which tracks people from their birth in 2000-2001 to adulthood through multiple waves. The study used a cluster-stratified framework, oversampling groups such as those from the smaller nations of the UK, those from ethnic minority groups, and those from disadvantaged areas. The entire dataset for the first wave comes from around 19,000 participants¹⁴. After dropouts, and after excluding cases in Scotland (where the exam system is different), cases with incomplete data on travel behaviour across ages 7, 11 and 14, and cases where travel mode was listed as 'other' or 'bike (someone else cycles)', we had records from 6778 participants who provided data on GCSE (General Certificate of Secondary Education) or iGCSE (international GCSE) exams at age 16 (5033 from England, 1023 from Wales, 722 from Northern Ireland; 3078 male, 3423 female, 276 missing data)."

Citation 14 has been updated to Joshi and Fitzsimons (2016) who describe the MCS reflecting the waves used in our study.

4. Please explain about GCSE exam because readers not familiar with the UK school system may not understand why the exam is relevant to be used as an outcome in this study.

The following clarification has been added to the Method: "GCSEs provide a useful metric as they are intended to assess a broad range of the subjects learned at school and a set of (typically) 7-9 GCSE assessments has been undertaken by almost everybody in England, Wales and Northern Ireland at age 16 since the mid-1980s."

5. Please elaborate more on the three mediators, for example how they are measured, and brief descriptions and validation information of the scales used to measure the three mediators rather than just the names of the scale.

We cite the two papers that describe the creation and validation of the standardized scales used here to measure the mediators: "We tested three potential mediators: Externalizing, which is a collective measure of inattentiveness, hyperactivity and conduct problems from the Strengths and Difficulties Questionnaire (SDQ)¹⁷; SDQ Internalizing, which measures emotional and peer-relationship difficulties¹⁷; and the Rosenberg Self-Esteem measure¹⁸." Readers wanting more detail will be served by those citations.

6. The UK Millennium Cohort sample is provided with weights so that it can be representative of each nation. However, it was not clear in the methods section that the sample is weighted and whether the statistical analysis took sample weighting into account.

We have now added the following sentence to the Method: “The Millennium Cohort contains weightings to enable descriptive statistics properly to represent the various countries within the UK, but these were not needed here given that we did not report simple descriptive statistics, and given country was included as a predictor in the analysis.”

Results

7. Please include a brief description of the sample included in the study (e.g., how many were girls and boys, how many belong in certain household quintiles, many were driven by car in year 7 and how many in year 14).

As requested, counts of male and female participants are now described in the Method section. New text has been added to the Method to provide the modal breakdown, as requested: “At age 7, travel to school was by the following modes: 3089 walking, 108 public transport, 198 school bus, 3329 private car and 54 bicycle. At age 14 the numbers were: 2462 walking, 1152 public transport, 1207 school bus, 1832 private car and 125 bicycle”

Discussion

8. Page 10, line 34: In here you state adequately powered sample, but it was not clear how the sampling in the UK Millennium Cohort was done. The strength of the sampling method was not clear as well. Please elaborate more on this.

The sample and how it was obtained is described in reference 14.

In response to Point 3 above we substantially expanded the description of the dataset in the Method section, which now reads: “...a UK birth cohort study which tracks people from their birth in 2000-2001 to adulthood through multiple waves. The study used a cluster-stratified framework, oversampling groups such as those from the smaller nations of the UK, those from ethnic minority groups, and those from disadvantaged areas” which we believe addresses the second point here concerning the strength of the sampling method.

Minor comments:

Abstract

9. The explanation about travel mode was significantly moderated by household income quintile was important and should be mentioned in the abstract.

The following sentence has now been added to the Abstract: “The relationship between travel mode and exam performance was moderated by household income quintile, most notably with poorer exam performance seen in high-income children who were driven to school”

Introduction

10. Page 5, Line 34: What are these important educational opportunities? If word limit allows, please elaborate.

The following has been added as clarification: “for example, obtaining parental advice on homework” (which is a specific example from the literature that we cited)

11. Is there any literature on benefit of walking together with parents to school? Because it seems that in the last paragraph of the introduction independent travel is pitched against escorted car travel with parents where interaction with parents becomes the latter’s example of benefit. I think this interaction can also happen where children walk together with their parent/s to school.

This is a good question. We are not aware of literature on this topic, although we agree that the mechanisms that have been proposed to offer an advantage to children driven to school by adults could also apply to children walked to school by adults. Unfortunately, as explained in the new paragraph we have added to the Method section, data on adult escorts was only collected in one wave of the study (at age 11). Presumably the MCS team assumed children were generally escorted

below this age and generally unescorted after it. Critically, as now explained in the paper, we had concerns about respondents misunderstanding this question (hundreds of 11 year olds were reported to be driving to school in private cars without any adults present [!]). Given these issues with the data, we are not able to provide analysis of this. But ultimately, given our conclusion is that travel mode makes very little difference to educational success, this question becomes moot anyway!

Methods

12. Page 7, line 34: I think creating a graph to show the temporal chains would be helpful to aid the readers understanding of them.

We can certainly do this, but worry that such a diagram would take up substantial page space without adding any further information over the chains already shown in the text. EDITOR: would you like this extra diagram?

13. Please describe how travel to school mode was assessed in the UK Millennium Cohort. From the Methods section, it is not clear how many different options of travel modes are available from the methods, even though later we could clearly see in the results table.

We have added the following text: "The original dataset uses numerical codes to describe the following categories: public transport, school bus/coach, car/taxi, bike, bike (someone else cycles), walking, or refusal/don't know/not applicable/other. We reduced these to: "Public transport", "School bus or coach", "Private motorised", "Bike" and "Walk", and excluded the other categories from analysis."

14. Why was walking used as the reference instead of being driven by car? I would assume the narrative would be easier if being driven by car was the reference.

A key reason we did not use the car as the reference category is that around 20% of UK households do not have a car. As such, to use the car as the reference category would arguably have been problematic unless we reduced the dataset to include only households who could have used a car. Walking is (nominally) an option open to all and so, we suggest, provides the more natural reference. In addition, across ages 11 and 14 walking was the predominant mode, further lending support to this decision.

Results

15. If possible, please create a visualisation for the results of the causal mediation analyses.

Presumably the request here is for Preacher & Hayes-style triangular X-M-Y diagrams? We could create these diagrams, but as it would require 6 diagrams to describe the results that are already reported – and far more to show the full set of analyses we conducted, also including the non-significant mediation effects – we fear they would add little information or nuance to the statistics already in the text. EDITOR: would you like these diagrams?

Discussion

16. Page 10, line 18: Was there any instance that some children that chose walking to school were walking together with their parents to school in the UK Millennium Cohort? If yes, it can be assumed that the UK Millennium Cohort was not able to capture this. How would this fit with the quality time with adults (in car trip to school) vs independent mobility?

Unfortunately, as explained above in reference to Point 11 (and now also explained in the ms), the data on adult escorts was clearly unreliable and so we are not able to answer any questions on this.

Conclusion

1. Page 11, line 18: "it is perhaps surprising they do not translate into more substantial academic performance advantages" – do you have a research recommendation to address this?

No, and it's not really clear how we'd research further the finding that there was no effect. Whatever cognitive skills go into (holistic) GCSE performance, these are apparently not notably affected by school travel experiences to the point they influence assessment outcomes.

Figure 1

2. Have you checked whether the palette used to create this graph is colour blind-friendly?

Thank you for this reminder. We have replaced the diagram with a version that has been checked using an online tester.

3. The lines with * and ** above the bars are confusing; it is not clear which part of the text in the results it is referring to.

The confusion was probably because the graph in our submitted ms was the wrong one! Somehow a version of the graph had been included which set motorised transport as the reference category on the left. Sorry - this is now corrected. The * and ** indicators now agree with the interaction terms described in the text and in the regression analysis table. NB The text always described the correct version - it was only the figure at the end of the ms that was wrong.

Reviewer: 2

Comments to the Author:

A well written and concise paper. Maybe a bit too concise. I would like to have some more details about the methods and some elaboration of strengths and limitations in the discussion.

More details about the method have been added in response to Reviewer 1, and we believe these will also satisfy Reviewer 2.

We are pleased to have the chance to address strengths and limitations and have added the following paragraph to the Discussion: "This study draws strength from its large, representative, high-quality longitudinal sample and its use of an official, national-standard set of examinations – taken by practically everybody in the three countries studied – as the outcome measure. Limitations attach to the self-report measures of travel mode (in particular we identified concerns with the quality of some of the data on adult escorts) and our method needing to reduce each child to a single mode, thereby likely missing nuances of children who have more complex school journeys, such as returning home using a different mode."

Importantly, the data (.i.e sample, survey instruments etc, study design) about travel mode to school, which after all is the main input variable to the analysis is not described. There is a reference to a secondary publication, where i suspect these might be better described. But that is not good enough. So please provide more info in this in the methods section.

Further details on the Millennium Cohort dataset have already been added in response to Reviewer 1.

The authors acknowledge that the way in which travel mode data is captured is a limitation, but this is only in the abstract. Please elaborate briefly on the major strengths and weaknesses in the discussion.

This has been addressed in our response to the first of this reviewer's comments.

I could not see any supplementary material, so i have not been able to ascertain whether this was sufficiently completed (as per Q13 in the review form).

Reviewer: 3

Comments to the Author:

Thanks for sending me this paper to review. It is an interesting read and I applaud the authors for submitting this for publication even in the absence of 'large' or 'positive' results. I think it is well conducted and written; I recommend it for publication and only have very minor comments
Thank you!

The abstract gives a nice sense of the paper. I would clarify here what scale these GCSE scores are measured in as most people are aware of these are being based on letter grades.

The abstract now specifies in more detail that the outcome measure is "School-leaver GCSE exam scores, each measured as a numerical score on a 0-9 scale and then summed to provide a single measure of educational success."

The introduction is useful and interesting. I was not aware of the research/claims that being driven to school is an educational opportunity (!). The wellbeing components are nicely set out

Methods

The sample size is a bit lower than the overall N in the MCS. The missingness across waves and variables could be better described.

When preparing the data, we almost immediately removed any cases that were not complete on all of the measures that we intended to use for analysis. We have added a little more detail to the paper which now says "The entire dataset for the first wave comes from around 19,000 participants¹⁴. After dropouts, and after excluding cases in Scotland (where the exam system is different), cases with incomplete data on travel behaviour across ages 7, 11 and 14, and cases where travel mode was listed as 'other' or 'bike (someone else cycles)', we had records from 6778 participants who also provided data on GCSE (General Certificate of Secondary Education) or iGCSE (international GCSE) exams at age 16 (5033 from England, 1023 from Wales, 722 from Northern Ireland; 3078 male, 3423 female, 276 missing data)". We hope this is sufficient.

The analyses are well motivated. One minor suggestion is about the exam score – could this perhaps be converted into standardised units or similar for use in the results and tables. This might help the reader have a sense as to the size of these associations

GCSEs were measured in letter grades (A-U) until recent years when they switched to the 0-9 scale that we used here. This means measuring performance on the 0-9 scale is arguably the 'proper' measure for the topic

https://assets.publishing.service.gov.uk/government/uploads/system/uploads/attachment_data/file/800507/GCSE_factsheet_for_parents__final_.pdf Specifically, working in these units provides regression coefficients that directly speak to real-world effect sizes in language that will make sense to parents, teachers and students, and so we believe is more informative than converting to Z scores or similar. For the convenience of people like ourselves who took GCSEs in the past, in the text we several times relate the numbers back to the old-style letter grades, e.g., "there were very clear cumulative advantages for children in higher-income households, peaking at a marginal mean difference of 24.46 points for the highest quintile versus the lowest (roughly equivalent to 8 exams each rising from grade C to grade A or to a child gaining three whole extra high-grade qualifications)". The part of this text underlined here was added to the paper in response to this reviewer's comment.

A little more detail on how this final model was chosen would be helpful beyond – the model simplification was based on theory, or on statistical significance?

The initial set of predictors was theoretically derived, but model simplification was purely statistical. This is now clarified in the paper.

The results and discussion do a nice job of setting out the main points and findings. There is not much on the limitations which could usefully be added

This was addressed in the responses to Reviewer 2.

Reviewer: 1

Competing interests of Reviewer: I declare no competing interests.

Reviewer: 2

Competing interests of Reviewer: No competing interests

Reviewer: 3

Competing interests of Reviewer: I am mentioned in the acknowledgements and had some e-mails with Ian Walker about the use of the MCS but otherwise have not seen this paper

VERSION 2 – REVIEW

REVIEWER	van Sluijs, Esther MRC Epidemiology Unit
REVIEW RETURNED	21-Feb-2023
GENERAL COMMENTS	The authors have clearly carefully considered the comments and made appropriate edits to the paper.
REVIEWER	Lavery, Anthony Imperial College London, Primary Care and Public Health
REVIEW RETURNED	27-Feb-2023
GENERAL COMMENTS	authors respond well to comments and this paper is suitable for publication